# Sustainable Antibiotic-Free Broiler Meat Production: Current Trends, Challenges, and Possibilities in a Developing Country Perspective

**DOI:** 10.3390/biology9110411

**Published:** 2020-11-23

**Authors:** Md. Hakimul Haque, Subir Sarker, Md. Shariful Islam, Md. Aminul Islam, Md. Rezaul Karim, Mohammad Enamul Hoque Kayesh, Muhammad J. A. Shiddiky, M. Sawkat Anwer

**Affiliations:** 1Department of Veterinary and Animal Sciences, Faculty of Agriculture, Rajshahi University, Rajshahi 6205, Bangladesh; msips06@ru.ac.bd; 2Department of Physiology Anatomy and Microbiology, School of Life Sciences, La Trobe University, Bundoora, VIC 3086, Australia; 3Department of Medicine, Faculty of Veterinary Medicine and Animal Science, Bangabandhu Sheikh Mujibur Rahman Agricultural University, Gazipur 1706, Bangladesh; aminulmd@gmail.com; 4Livestock Research Institute, Savar, Dhaka 1341, Bangladesh; rezavetmicro@yahoo.com; 5Department of Microbiology and Public Health, Faculty of Animal Science and Veterinary Medicine, Patuakhali Science and Technology University, Barishal 8210, Bangladesh; mehkayesh@pstu.ac.bd; 6School of Environment and Science (ESC), and Queensland Micro- and Nanotechnology Centre (QMNC), Griffith University, Nathan Campus, 170 Kessels Road, QLD 4111, Australia; m.shiddiky@griffith.edu.au; 7Department of Biomedical Sciences, Cummings School of Veterinary Medicine, Tufts University, 200 Westboro Road, North Grafton, MA 01536, USA; Sawkat.Anwer@tufts.edu

**Keywords:** sustainable agricultural system, antibiotics, antibiotic-free, antibiotic resistance, broiler meat production

## Abstract

**Simple Summary:**

Chickens are raised with the assistance of the regular use of antibiotics, not only for the prevention and treatment of diseases but, also, for body growth. Overuse and misuse of antibiotics in animals are contributing to the rising threat of antibiotic resistance. Therefore, antibiotic-free broiler meat production is becoming increasingly popular worldwide to meet consumer demand. However, numerous challenges need to be overcome in producing antibiotic-free broiler meat by adopting suitable strategies regarding food safety and chicken welfare issues. This review focuses on the current scenario of antibiotic use, prospects, and challenges in sustainable antibiotic-free broiler meat production. We also discuss the needs and challenges of antibiotic alternatives and provide a future perspective on antibiotic-free broiler meat production.

**Abstract:**

Antibiotic-free broiler meat production is becoming increasingly popular worldwide due to consumer perception that it is superior to conventional broiler meat. Globally, broiler farming impacts the income generation of low-income households, helping to alleviate poverty and secure food in the countryside and in semi-municipal societies. For decades, antibiotics have been utilized in the poultry industry to prevent and treat diseases and promote growth. This practice contributes to the development of drug-resistant bacteria in livestock, including poultry, and humans through the food chain, posing a global public health threat. Additionally, consumer demand for antibiotic-free broiler meat is increasing. However, there are many challenges that need to be overcome by adopting suitable strategies to produce antibiotic-free broiler meat with regards to food safety and chicken welfare issues. Herein, we focus on the importance and current scenario of antibiotic use, prospects, and challenges in the production of sustainable antibiotic-free broiler meat, emphasizing broiler farming in the context of Bangladesh. Moreover, we also discuss the need for and challenges of antibiotic alternatives and provide a future outlook for antibiotic-free broiler meat production.

## 1. Introduction

There is a worldwide movement to support a sustainable agricultural system that involves sustaining farmers, resources, and societies. It is expected that this can be achieved by upgrading farming practices that are profitable, environmentally sound, good for communities, and antibiotic-free. However, raising animals without antibiotics is a challenge, and especially, antibiotic-free broiler meat production is a growing challenge in many developing countries, including Bangladesh, where antibiotics are used injudiciously [1]. Broiler chickens are reared particularly for meat production because of their typical soft, tender meat, low-fat content, and short production period. Broilers take the place of valuable food animals globally by notably contributing to food security, protein supply, and people’s employment [2]. Sustainable broiler production not only requires maximum productivity but, also, includes bird and human welfare and environmental protection. In addition, consumers are concerned about antibiotic residue and antimicrobial resistance, as well as pesticide residues, additives, nutritional content, flavor, traceability, regional production, genetically modified organisms, social justice, etc., with regards to broiler meat production. Therefore, broiler meat production without using antibiotics is crucial in the battle of antimicrobial resistance to save poultry, animals, and public health.

Broiler meat production has undergone exponential growth for global consumption and business profit. Low production costs and rapid economic progress are pivotal in its expansion [3]. Though commercial poultry farming is a profitable business, it faces a number of challenges. Among them, the occurrence of infectious and noninfectious diseases is a major challenge due to poor biosecurity and husbandry practices. Over several decades, some antibiotics have been used in broiler feed to control, prevent, and treat diseases and boost performance and feed efficacy [4,5]. This inappropriate antibacterial use favors antimicrobial resistance. The most common pathway for bacteria to gain resistance is through mobile genetic components, including bacteriophages, plasmids, naked DNA, or transposons. Plasmid-mediated gene transfer facilitates the flow of resistant genes between bacteria, accelerating antibiotic resistance [6]. Bacteria can also develop antibiotic resistance through sequential mutations in the chromosome, as happens in fluoroquinolone resistance. Using normal genetic variations, bacterial populations mutate to render antibiotics ineffective against them [7]. Indeed, antibiotic resistance begins with the interaction between bacteria and antibiotic, leading to the removal of sensitive bacteria and selection of resistant populations. However, the distribution and assortment of resistance are complicated matters that pose a severe public health problem [8]. The indiscriminant use of antibiotics in nonbacterial infections like influenza and other viral infections facilitates antibiotic resistance [9].

In Bangladesh, commercial poultry farmers extensively utilize antibiotics without any veterinary advice and often do not follow withdrawal period guidelines [10]. A lack of both easily accessible veterinary facilities and adequate knowledge combined with a high-profit motive are some of the factors that drive local producers to inappropriate and, at times, illegal use of antimicrobial agents [11]. The cost-effective production of broilers may act as a driver for the practice of using growth-promoting medicines, including antibiotics, for the overall growth performance [12,13]. These agents get stored in various body parts and tissues, resulting in antibiotic residues for unintended exposure to consumers. Sattar et al. [14] reported antibiotic residues mostly in the liver, kidney, thigh meat, and breast meat of broilers. Easy access to antibiotics from registered veterinarians by farmers without prescriptions contributes to the indiscriminant use of antibiotics, which can result in antimicrobial resistance from food animals [11]. Although antibiotic resistance has always been present in nature, antibiotic pressure due to inappropriate use facilitated the rapid and widespread emergence of drug-resistant pathogenic bacteria [15]. As several antibiotics frequently consumed by livestock are identical to those used in humans, there is worldwide fear that antibiotic-resistant bacteria may be transferred from animals to humans, leading to severe public health concerns [16]. Resistant bacteria with resistant genes can quickly spread among humans, animals, and the ecological community. Therefore, new approaches should be considered for antibiotic-free broiler production so they can be sustainably raised and marketed. Therefore, this review highlights the current scenario of antibiotic use in broiler meat production, as well as the global challenges, prospects, and approaches of antibiotic-free production, particularly in the context of Bangladesh.

## 2. Economic and Clinical Importance of Antibiotic-Free Broiler Meat Production

The fast growth of the poultry industry has been attained through the effective implementation of high-yielding strains of meat-type chickens and the availability of nutritionally balanced feed. Broiler farming plays a central role in enhancing income, improving food safety, and mitigating scarcity in the countryside and in semi-urban communities in developing countries. For a couple of decades, the poultry industry has played a critical role in the economic development of Bangladesh by providing job opportunities, food security, and good-quality protein. The poultry industry has assisted in changing living and food habits by moderating animal protein sources such as beef and mutton. For instance, about 44% of the protein for everyday human consumption comes from poultry and animal products [17]. As a primary portion of production, this industry’s ability to furnish the country with affordable and nourishing protein in the form of meat and eggs is encouraging [18]. There are over a million small and large poultry farms, 80 grandparent (GP) stock farms, 130 parent stock farms and hatcheries, and over 50 feed mills across the country, making up a USD 2.36 billion industry. About five million people are employed in the industry directly and indirectly. A study estimated that around 150,000 poultry farms in Bangladesh are producing 570 million tons of meat and 7.334 billion eggs annually, and about 68.17% of animal protein consumed comes from poultry meat [19]. A significant number of pastoral individuals depend on this business for their income. The poultry industry has made up one-third of the entire agronomic contribution to the Gross Domestic Product (GDP) (18.60%) in recent years [20].

In food animal production, antibiotics are also exploited, albeit unwisely, to advance animal growth and feed efficiency [21,22]. Generally, farmers in developing countries use antibiotics deliberately to promote growth without any veterinary consultation. Additionally, shockingly, 88% of producers did not adhere to the required antibiotic withdrawal time before marketing [22]. On the other hand, only 10% of farmers stopped using antibiotics before marketing, and only 2% of farmers withdrew antibiotics at least seven days before marketing. Still, they did not always follow this withdrawal protocol [23]. Generally, excessive and unnecessary use of medicines affects the total cost of production. A study on the antibiotic usage patterns in Bangladesh concluded that the use of antibiotics has an economic effect on broiler production [11]. Another study revealed that the cost of antibiotic usage could be 3.53% of the overall production rate based on responses from 84% of farmers [23]. The cost of antibiotics varies from farm to farm due to the presence and severity of diseases and their use as growth promoters. Removing antibiotics from broilers can lower the production cost and, thereby, decrease the market price of broiler meat.

The nontherapeutic use of antibiotics increases the residual accumulation more than medically necessary to use and, thereby, increases the incidence of antimicrobial resistance. The consumption of residual medicines through animal food products (e.g., meat, milk) is assumed to initiate resistance development in humans. Besides, commensal bacteria of livestock are often present in fresh meat, which can serve as reservoirs of resistant genes to be relocated to pathogenic bacteria in humans [7]. Studies have shown that antibiotic residues were present in more than 50% of samples of broiler and layer meat collected in different regions of Bangladesh [14,22]. Another study by Faiz and Bashe (2011) reported that some pathogenic bacteria, such as *Pseudomonas aeruginosa* and *Salmonella typhi*, are resistant to commonly used antibiotics [24]. The high degree of antibiotics resistance in Bangladesh poses a regional and global threat [25]. In Bangladesh, diarrhea accounts for around 230,000 child deaths annually, and a significant portion is due to antibiotic-resistant bacterial infection [19].

## 3. Prospects of Antibiotic-Free Broiler Meat Production

In veterinary practice, antibiotics are used for prophylactic, therapeutic, and growth-promoting purposes. The purchase of medicines without a prescription is common in developing countries like Bangladesh. This can lead to an inadequate course of treatment, incorrect antibiotic selection, indiscriminant and excessive use, and enhanced residual deposition in animal bodies. A recent study with 160 poultry samples from different regions in Bangladesh identified antibiotics in more than 50% of the samples [22]. Producing a sufficient amount of food for a growing population is a great challenge for any country. Of the 17 Sustainable Development Goals (SDGs), 14 are connected to the poultry sector in some way or other. According to data from the Bangladesh Poultry Industries Central Council (BPICC), the annual commercial production of eggs and poultry meat is about to 1022 crore pieces and 14.6 lakh tons, respectively. Thus, the poultry industry appears to be a big sector, worth about USD 353.88 crore [26]. Nowadays, consumers are refusing to consume antibiotic-treated chicken and demanding safe, antibiotic-free beef. Therefore, there is a large market for antibiotic-free chicken in Bangladesh. The challenge is to find a way to produce them. One option would be to use medicinal plants instead of antibiotics as growth promoters. Bangladesh is rich in medicinal plants, and many such plants have been shown to have antibacterial activities, such as *Abutilon indicum* L., *Caesalpinia bonduc* L., *Ixora nigricans* L., etc. Many natural foods also have antibacterial actions, such as broccoli (*Brassica oleracea*), guava (*Psidium guajava*), garlic (*Allium sativum*), and okra (*Abelmoschus esculentus*) [27,28,29,30]. Based on antibacterial medicinal plants and natural foods, a safe antibacterial preparation could be developed to use in food animals as an alternative to antibiotics for growth promotion and feeding efficiency.

## 4. Significant Challenges for Sustainable Antibiotic-Free Broilers Production

Bangladesh is a developing country, and its seasonal variations are very distinct, categorized by extreme temperatures, excessive humidity, and heavy rainfall. The environment is beneficial for the survival and growth of various microorganisms and makes them ubiquitous and fastidious. Most poultry farmers are not well-educated or well-trained for modern scientific farming. It is estimated that about 80% of farmers lack adequate knowledge about rearing current high-yielding strains [23]. Moreover, farmers in Bangladesh can quickly get antibiotics from local vendors without a veterinarian’s prescription. Biosecurity measures are also terrible; thus, the entry of microorganisms on most farms is common, affecting the production performance of flocks. The commercial livestock sector is multiplying and conveying a lot of diseases with it. Thus, biosecurity must be implemented without fail [26]. Newcastle disease, Gumboro or infectious bursal disease, Marek’s disease, duck plague, fowl pox, fowl cholera, leucosis, and infectious bronchitis are among the most common infectious diseases in the poultry sector [26]. It is therefore essential to control primary and secondary infections for successful poultry farming in Bangladesh. One of the most critical factors in antibiotic-free broiler production is the economic challenge, i.e., cost management. The most critical challenges in producing antibiotic-free broilers in Bangladesh and abroad (Figure 1) are addressed below.

### 4.1. Environment

Whatever system is followed in any corner of the world, none can deny that environmental pollution is a part of broiler production. Disease and mortality are likely to increase due to polluted and hazardous environments. Increased clinical and subclinical necrotic enteritis (NE) and related disorders, including cholangiohepatitis, on poultry farms are the likely consequences. Therefore, it is vital to keep the level of pollution to a minimum for the well-being of both birds and humans. To achieve this goal, farmers should prioritize proper carcass disposal and waste management during farm operations. While about 70% of all agricultural land globally is occupied by livestock [31], poultry farming’s effect on agricultural land loss is minimal [32]. However, another study suggested that feed production, processing, and transport result in a greater overall environmental impact than any other part of broiler production [33,34]. Therefore, farmers must follow the principles of poultry production, upholding public interest, as well as a hazard-free environment, thereby ensuring sustainable antibiotic-free poultry meat production.

### 4.2. Resource Management

Resource management is the process by which farm owners effectively utilize their tangible and intangible resources. It involves effective planning to ensure that the right resources are assigned to the right tasks. Broiler meat production without the use of antibiotics requires farmers to decrease the stocking density, raise the downtime, clean rapidly and frequently, maintain the correct temperature, provide rigorous biosecurity, reduce hassle, select the best breed, etc. Therefore, it is a significant undertaking for the efficient utilization of limited resources in broiler farming management. Broiler production and meat processing require an adequate supply of water. Thus, inadequate water availability is a significant obstacle to sustainable antibiotic-free broiler meat production [35]. Another relevant aspect of resource management is feeding the poultry. There is competition between poultry and humans for quality proteins. This creates a challenge when a protein-rich foodstuff (wheat and maize) needs to be included in the supplied ration [36,37].

### 4.3. Antibiotics

Despite increasing concerns, antibiotics are used worldwide on poultry farms, particularly broiler farms, to treat and prevent diseases and as growth promoters to improve meat production [38,39,40]. Antibiotics are commonly used to treat intestinal infections caused by *Salmonella*, *Escherichia coli* (E. coli), or *Clostridium* spp. These pathogens can negatively impact poultry farming and farmers’ profit. A recent study demonstrated the patterns of antibiotic use at selected broiler farms in Bangladesh, with reported use for therapy (43.8%), prevention (31.5%), or both (47.9%), and for promoting growth (8.2%) [11]. However, antibiotic use varies from nation to nation depending on several factors, including knowledge, economy, management system, and farming practices [41]. Although the monitoring of antibiotic use is essential, it is poorly implemented around the world, including in Bangladesh [40]. According to a recent estimate, the average annual global consumption of antimicrobials per kilogram of chicken production was 148 mg [42]. Furthermore, a rise of the antimicrobial consumption by 67% was projected [42]. The widespread use of antibiotics on broiler farms could encourage antibiotic resistance, ultimately causing a global public health threat [43]. Alternative approaches should be investigated to replace growth-promoting antibiotics to maintain and ensure gut health and good-quality meat production. However, producing quality broiler meat at consumer-friendly prices without antibiotics will be a challenge.

### 4.4. Breed Diversity

In the modern period, it is evident that there is minimal breed and genetic variability among commercial birds. Researchers have found that the breed or genetic diversity of commercial chicken farms has declined in the developed world. This is mainly because modern industrialized agriculture depends, in no small degree, on uniformity. There are hundreds of breeds of poultry that are well-recognized and registered. Most of these poultry breeds and varieties are locally adapted for multiple purposes and suited to various farming enterprises. They are mostly reared under small-hold production systems and are a large part of poultry meat production worldwide. On the contrary, modern chicken strains are raised for either meat or eggs, and only limited high-producer strains are employed. However, such top-yielding hybrids are more susceptible because of their compromised immune capacity [44]. Not cultivating other breeds means not only sacrificing the natural genetic diversity but, also, decreasing the birds’ potential to adapt to new conditions and circumstances, such as climate changes [45]. It is noteworthy that genetic uniformity or a lack of genetic diversity makes birds more vulnerable to many diseases. While it is convenient and profitable to mass-produce a particular breed, the loss of diversity makes it more challenging to adapt to changes [46].

### 4.5. Social Aspects

Two social issues, working environments and human health, and the impartiality of job agreements, including health care, fair wages, and cooperative trading rights, are significant matters in sustainable broiler production. Poultry farmers and their family members risk becoming contaminated with antibiotic-resistant bacteria and developing antibiotic allergies. Health ailments in farm workers due to exposure to dust, litter, and feathers are quite common [47] and should receive attention to assure production continuity. According to Quandt et al. [48], “chicken grabbing” in the USA is characterized by a working atmosphere and work organization that encourages accidents and illness and is closely connected to the risk of disease, human rights, and overall working environments. Workers are exploited disproportionately in food-producing industries such as the poultry sector. For example, healthy working conditions are not maintained in order to keep the direct costs of production low. Besides, the terms and conditions with contract growers vary greatly and, in some cases, are grossly unreasonable [49,50].

### 4.6. Gender Issue

Among the 17 Sustainable Development Goals established by the United Nations in 2015, promoting gender egalitarianism and empowering women is noteworthy. Commercial poultry production on an industrial scale does not firmly follow this gender goal. However, small-scale family poultry production with low input is mostly run by women in most Asian and African countries. Such a production system supplies much-needed poultry meat and eggs for the proper nutrition of millions of people worldwide and helps reduce poverty and improve incomes in rural areas. Dolberg et al. [51] reported that women are actively involved in various aspects of small-scale poultry production in rural households. While women are often responsible for taking care of poultry, they usually face more critical challenges than men. Women have less access to and influence over property, credit, labor, technology, and the facilities required to take advantage of development opportunities. In some instances, they do not have full control or decision-making authority over the birds or the revenue from their sale [52]. Further, the keeping of family poultry (FP) is typically the responsibility of women. Unfortunately, there is a lack of knowledge of women’s and men’s particular roles in FP production; this results in gender blindness. Therefore, the contribution of women to this subsector of poultry remains invisible [53]. Thus, the gender issue is a massive challenge in sustainable antibiotic-free broiler meat production in Bangladesh.

### 4.7. Animal Welfare

It is imperative that we look after the welfare of animals that serve us in so many ways. Humanity demands it. With the advancement of time, the animal welfare issue has become a matter of great concern. The suffering of poultry caused intentionally or unintentionally cannot be justified in any production [54]. The Animal Welfare Standards research project funded by the EU laid down 12 standards for animal protection. For example, poultry should not be stressed from excessive hunger or thirst, should have room for free movement, and should show usual nonharmful behavior. Poultry producers should promote positive emotions and avoid negative emotions, such as fear, distress, frustration, and apathy. These criteria are currently not fulfilled by many producers, and some poultry production systems may cause suffering [55]. Skipping a day of feeding broiler breeders to control their body weight affects the chickens’ welfare under the modern husbandry system. This practice needs the attention of researchers to ensure the sustainability of production in the future. It is interesting to note that some studies recommend providing low-density feeds to increase female broiler breeders as an alternative to feeding constraints [56]. Any improvement in animal welfare also upgrades the sustainability and is less harmful to the environment. Nevertheless, it is challenging to balance the relationship between poultry production, animal welfare, and the impact of production on the ground. Organic animal production is often set in with sustainability, which can lead consumers to think that animals raised organically have better welfare. Although some organic poultry systems uphold high levels of interest, it can be challenging to maintain good health and well-being on organic poultry farms [57].

### 4.8. Institutional Aspects

The institutional dimensions of sustainability comprise the management and governance of global processes and ensure that organizations are responsible, accountable, and transparent to their members and representatives. Indeed, a few breeding corporations dominate a significant part of the industry, making it hard for small businesses to start up. The producer should balance the multiple interests inherent in the principle of sustainability with the management of operations. Although Bangladesh’s government has many policies to meet the SDGs to tackle the economic, environmental, and social aspects, there is still no poultry policy to coordinate with other enterprises. One policy in a field often hampers advancement in another. Solutions to difficult problems are sometimes in the hands of policymakers in different industries or at various government levels. It is a significant cause of many unsustainable long-term patterns [46]. The sustainability of poultry production involves producing sufficient outputs to provide satisfactory products for consumers and income for farmers and constant availability of the required inputs such as raw materials, labor, and capital in the long term. On a biological basis, the production of village-based poultry in Bangladesh is almost sustainable. Almost all of the necessary feed is found through scavenging, and home-grown grain and byproducts serve as feed supplements. Although poultry meat and egg productivity has increased continuously during the last decades, it is questionable whether the consequent rapidly rising need for poultry products will be fulfilled in the future. The challenges of small-scale and comprehensive poultry production have been discussed previously [58]. In recent years, Bangladesh’s action plan to train vast numbers of rural extension staff and coordinate the supply of vaccines at the rural level has been useful. There is also a need to ensure the long-term viability of the organizational structure and training programs.

### 4.9. Consumption Pattern and Demand for Poultry Products

The requirements for poultry products will be doubled in 2050 based on the rising usage in recent decades [49]. It is believed that today’s food insecurity problems are not due to a shortage of food but, rather, to poor people’s ability to afford it. Increasing consumption patterns in some countries will demand increased production. In contrast, as people grow wealthier, the demand for healthier “green” diets or organic foods will increase in some other countries [46]. On the other hand, the indiscriminate use of antibiotics or other harmful substances in the production chain negatively affects consumers’ trust in broiler meat. These results negatively impact market demand, as observed recently in Bangladesh. A lack of confidence in the product threatens sustainable production, causing colossal loss to the industry. Farmers must balance order, supply, and consumption patterns in a transparent and healthy competitive environment. Further, the quality of feeds and feeding programs is also a component of sustainable broiler production. The purpose of a feeding program is to feed the chickens and not the bacteria. A successful feeding program is essential in antibiotic-free broiler production. Poor-quality feed ingredients, rancid animal byproducts in the feed, and mycotoxin contamination can damage gut health and the intestinal epithelium. Therefore, excellent feed ingredients are essential to keep proper gut health, as well as better flock health. However, these ingredients will increase the production cost unless the savings from not using antibiotics can offset the cost of quality feed. This is an important consideration, since consumers in Bangladesh are demanding high-quality products but are unwilling to pay more. Therefore, it may be a significant challenge to supply antibiotic-free broiler meat at a price desired by consumers.

## 5. Possibilities of Antibiotic-Free Broiler Production

Superior output and biosecurity management practices are essential components in raising chicken without antibiotics. Some general approaches for antibiotic-free broiler production are shown in Table 1 and discussed below.

### 5.1. Strict Downtime between Placements

It is a quick, simple management procedure to raise antibiotic-free broilers. In this case, new flock entry should be restricted to a minimum of 14 days (downtime) with good cleanout (clearing out birds and removing the litter cake). The downtime should start when birds are out of the house production site.

### 5.2. Optimum Stocking Density

The stocking density is a crucial concern for profitable broiler farming. As the stocking density increases, so does the stress level of birds. Increased stress is associated with compromised immunity and, consequently, an increased susceptibility to disease. Thus, decreasing stress by increasing the stock density can help reduce the risk of disease and the need for antibiotics. A reduced stocking density is also an excellent option to help keep litter moisture at a minimum compared to the current conventional program. Farmers should decrease the stocking density, which, in turn, reduces the shedding of cocci oocyst and pathogenic bacteria. This practice also helps in proper ventilation and aeration and, thus, reduces stress. However, the ideal stock density is still being debated. Bilgili and Hess [59] reported a significant improvement of the overall growth performance, carcass quality, and mortality with a decreased stock density. In Bangladesh, usually 1- to 1.5-square-feet of floor space is assigned to each bird. However, in antibiotic-free broiler production, providing more space than required is an important factor.

### 5.3. Good Litter Management

Keeping the litter dry is critical in the overall management of any poultry farm, as it influences the bird performance by decreasing the ammonia level and, thereby, the growers’ profits. Dry waste and low ammonia levels are the keys to success in raising poultry. The negative impact of ammonia on bird performance is well-documented [60,61]. Dry litter controls the ammonia levels and provides a better flock atmosphere and reduces disapproval due to contact dermatitis and inflamed bursa. When unused bedding materials start to hold moisture, it will clump together to form caking, leading to increased ammonia formation. Caking is mainly due to moisture in the litter and can be prevented by having adequate ventilation. Broilers drink approximately two pounds of water for every pound of feed they consume. About 20% of the water is used for growth, while most of it eventually reaches the litter as waste. Therefore, there must be adequate ventilation to remove moisture within the waste to prevent caking. Generally overventilation is needed to fix the problem once caking has started. Growers should attend to litter sanitation and treatment at regular intervals. It is a great idea to monitor the ammonia levels regularly, and handheld sensors can accurately manage them. Amending the litter and providing adequate ventilation are the most practical ways to control ammonia and improve litter quality [62]. Preventive fan maintenance can aid in keeping the ventilation program effective. A total litter cleanout at least once a year is highly recommended.

### 5.4. Control Environment Housing

Producers can reduce bacterial growth and antibiotic use by maintaining the adequate ventilation of poultry housing facilities. A study reported the presence of antibiotic-resistant bacteria in the air of broiler chicken facilities [63]. Dust is an excellent carrier of bacteria, while mold is a potential source of infection. For appropriate airing, heat, relative moisture, and light, an environmentally controlled structure should be built oriented in the east-to-west direction longitudinally, with large exhaust fans on the west side and evaporative chilling plugs on the east side, accompanied by programmed feeding and drinking methods inside [64]. Chick suppliers can provide useful information on the ideal temperature, humidity, and air quality for raising broilers. Airborne dispersion of antibiotic-resistant bacteria should not be underestimated, considering the respiratory health of chickens and poultry workers. Farmers should follow the standard guidelines for maintaining adequate ventilation [64].

### 5.5. Pre-Starter Feed

Pre-starter feed is crucial in establishing good gut microbe populations. To achieve a balance in gut microbiota for nutritional enhancement of the gastrointestinal (GI) tract, boost the immune development, and facilitate nutrient absorption, it is useful to include probiotics, prebiotics, organic acids, short- and medium-chain fatty acids, and phytogenic feed. The broiler industry has achieved noteworthy advances in productivity by prioritizing the hereditary selection, diet, and controlling procedures. The nutrition for broilers is focused on the need to get to the market weight as quickly as possible. There is a growing interest in a specifically designed pre-starter feed for 10–14 days of broiler growth, since this represents more than 20% of the growth time [65]. A specialized formulation should ensure that the feed is easily digestible and meets the broiler’s nutritional needs. The efficacy of pre-starter diets relies on the potential carryover effect to increase the bird performance until market age. The nutrient requirements, particularly digestible amino acids, are crucial in the broiler’s first 10 days. Rice and its lower non-starch polysaccharide substance may be a better option than corn or wheat for pre-starter diets. Additionally, the addition of fibrous components in low-fiber feeds at earlier stages may enhance the gut development and improve nutrient absorption, eventually enhancing the overall growth performance. Finally, the manufacturer should concentrate on feed formulations to remove antibiotic growth promoters from all broiler diets. This will ensure sustainable growth and gain consumer acceptance. Broiler feed should receive particular attention by providing the particle size, protein and fat qualities, and specific additives.

### 5.6. Water Quality and Sanitation

A good water sanitation program is key to healthy poultry production. Water is essential for birds. A routine water quality analysis for bacteria, pH, hardness, minerals, and total dissolved solids should be conducted annually. Flushing and disinfecting between flocks with suitable quality disinfectants should be done to remove biofilm from the pipeline. Biofilm is a sticky film that can exist inside water lines, regulators, and nipple drinkers and can be composed of bacteria and other organisms [66]. Chlorine is the most popular sanitizer; other well-documented sanitizers include hydrogen peroxide, chlorine dioxide, and ozone. In addition, acidifiers can improve the sanitizer’s effectiveness and reduce bacterial growth in water lines [67].

### 5.7. Antibiotic Alternatives

Over the past two decades, producers have extensively used antibiotics for broiler weight gain. Including antibiotics in broiler production enhances the food conversion ratio by 4% [68]. Thus, measures need to be in place to offset this loss and any undesirable growth performance in immunocompromised broilers [68]. Additionally, withdrawing antibiotic feed additives could raise necrotic enteritis in the flocks. Although removing antibiotic feed additives from the diet would enhance the production cost, this increased cost can be recovered from the expected increase in broiler meat prices [69]. It should be noted that the effect on broiler growth and weight from the withdrawal of growth-promoting medicines may occur only after the first year [70]. Thus, the campaign for an alternative to antibiotics should focus on antimicrobial stewardship [71]. These alternatives should provide a low mortality rate and an adequate level of yield while preserving the environment and consumer health. Much research has been conducted to explore natural agents with beneficial effects similar to growth-promoting antibiotics. In addition, several nontherapeutic alternatives can be substituted for antibiotics. The most popular options are probiotics, prebiotics, enzymes, organic acids, immunostimulants, bacteriocins, bacteriophages, phytogenic feed additives, nanoparticles, and essential oils (Table 2), and these alternatives will be discussed in detail below.

#### 5.7.1. Probiotics and Prebiotics

Probiotics are yeasts and live bacteria that confer beneficial effects on the health if administered in adequate doses (WHO, 2001). Probiotics can replace antibiotics by changing the intestinal microbiome, thereby producing some of the effects of antibiotics. For example, feed supplementation with probiotics improves the feed efficiency and intestinal health and, ultimately, facilitates the faster growth of broilers by reducing the intestinal pH, altering the intestinal bacterial composition, and improving digestive activity [72,73]. Probiotics stimulate endogenous enzyme production, which reduces the production of toxic substances and increases vitamins and/or antimicrobials such as bacteriocins [74]. It has been reported that bacteriocins inhibit the production of toxins and the adhesion of pathogenic microbes [75]. Probiotics decrease colonization by intestinal bacteria and, ultimately, stimulate the immune response [74]. The administration of *Enterococcus faecium*, *Streptomyces* spp., and *Bacillus* spp. in chicken feed triggers antibacterial effects on other pathogenic bacteria in the small intestine (Figure 2) [76,77,78]. Broiler feed supplemented with *B. subtilis* increased the body weights by 4.4% [78]. The H_2_S and NH_3_ concentrations in chicken excretions were also reduced after treating chickens with probiotics, leading to less odor. Probiotics increased the meat quality of poultry by affecting the fat and protein contents [74,79]. They are essential in improving the water-holding capacity, color, pH, oxidation stability, and the chemical composition of meat, such as the fatty acid content [79]. Feed supplemented with *B. licheniformis* improves the juiciness, flavor, and color of broiler chicken meat [80], which is appreciated by consumers [81]. Probiotic supplementation also lessens parasitic infestation in chickens [73]. Probiotics exert coccidiostatic effects on Eimeria tenella, maintain intestinal health, and reduce the spread and risk of coccidiosis in broiler production systems [81].

The nondigestible components of feed that exert potential beneficial effects on the health are known as prebiotics. Most prebiotics are fermentation products consisting of oligosaccharides and short-chain polysaccharides [82]. Their fermentable properties stimulate the growth and activity of beneficial bacteria in the ileum and cecum and contribute to a healthy intestinal tract and, ultimately, increase poultry productivity [83,84]. They are also considered as excellent alternatives to antibiotics. Prebiotics alter cecal proteobacteria composition and increase broiler growth [85]. Products containing high levels of mannose and mannoprotein in the feed tend to increase villus cell numbers in the intestines [86] and are essential for improving intestinal health, thereby producing healthier poultry. Adding 0.2% mannan oligosaccharides to the chicken diet improves the intestinal health more than antibiotics by reducing harmful pathogenic bacteria and increasing potentially beneficial bacteria [87].

#### 5.7.2. Organic Acids

Organic acids are conservation agents widely known for protecting feed from microbial spoilage and improving the nutrient digestibility in poultry [88]. Organic acids include carboxylic acids that carry a hydroxyl group on the alpha-carbon, such as malic, lactic, and tartaric acids, and pure monocarboxylic acids, including acetic, formic, butyric, and propionic acids. Organic acids can inhibit the microbial growth by disrupting bacterial enzymatic reactions and decreasing the transport of acidic compounds by nonionic diffusion through the membrane [89]. It has been reported that adding organic acids to feed may improve the growth, feed conversion rate, and feed utilization of broilers [90,91,92]. While drinking water is a risk factor for spreading campylobacter infection in broilers, Chaveerach et al. demonstrated that organic acid-treated drinking water can potentially prevent campylobacter infection in broiler flocks without any damage to gut epithelial cells [93]. Blends of formic and propionic acids in drinking water for chickens can generate homogeneous and distinct populations in the intestinal microbiota and increase *Lactobacillus* spp. colonization in the ileum [91], which can be a substitute for antibiotics used to reduce pathogenic bacteria in the gastrointestinal tract (GIT). These changes in the intestinal microbiota and increased *Lactobacillus* populations suggest that organic acids can substitute for antibiotics such as bacitracin to reduce pathogenic bacteria in the GIT [91]. Additionally, organic acids have the potential to inhibit *E. coli* infection [94], and a supplementation with 2% citric acid can improve the gut health [95]. It has been reported that formic acid, an extensively studied organic acid, can limit *Salmonella* infection and other foodborne pathogens when used in a poultry diet [96]. Several organic acids may also play beneficial roles in digestion. For example, when butyric acid is used as a feed additive, it improves the digestibility of ileal proteins from poorly digestible protein sources [92]. Butyric acid, a saturated carboxylic acid, is produced in the cecum and colon by fermenting carbohydrates such as dietary fiber and unabsorbed starch [97]. Moreover, butyric acid is a readily available energy source for intestinal epithelial cells and stimulates their multiplication and differentiation, thereby improving the feed efficiency and growth performance of chickens [82,92,98].

#### 5.7.3. Amino Acids and Enzymes

Broiler feeds are supplemented with amino acids and enzymes to increase the feed conversion [81]. These enzymes are produced by fermenting fungi and bacteria. Enzymes enhance the digestibility of the feed by facilitating the degradation of proteins, phytates, and glucan. For example, it has been reported that broiler diets of wheat and barley supplemented with endo-b-1-4-xylanases and b-1-3, 1-4-glucanases can improve the digestibility of the feed [99]. Another study reported that a phytase enzyme improved daily weight gain by increasing the villus width and decreasing the crypt depth [95]. Lysins are bacteriophage endolysins that could be an alternative therapeutic option instead of antibiotics. Lysins are phage-encoded peptidoglycan hydrolases that cause lysis of bacteria by cleaving peptidoglycan when applied exogenously to Gram-positive bacteria [100,101]. The antibacterial efficacy of several lysins has been reported; for example, peptidases, amidases, and lysozymes showed antimicrobial potential against *C. perfringens* in poultry [102].

#### 5.7.4. Phytogenic Feed Additives

Phytogenic feed additives (PFAs), also known as phytobiotics or botanicals, are derived from plants, herbs, and spices and can improve animal health. PFAs are reported to positively affect growth by improving the feed conversion ratio (FCR), boosting the immune system, and reducing stress. Several recent studies also showed that phytogenic feed additives promoted broiler chicken growth and could be used as an alternative to antibiotics [72,103,104,105,106]. Another study demonstrated that including *Lippia javanica* in broiler feed at 5 g/kg enhanced the broilers’ daily gain and slaughter weight. Phytogenic extracts of *Lippia javanica* leaf can stimulate glycolysis and increase energy production utilization and, thus, growth [107]. They can also improve the fatty acid profile of broiler chicken meat [107]. It has been reported that the use of garlic (5 g/kg) and black pepper powder (1 g/kg) in broiler feed could enhance broiler chickens’ weight gain and consumption index [108]. Thus, PFAs have the potential to replace antibiotics in the poultry industry. On the other hand, there are also essential oils, the hydrophobic concentrated liquid of odoriferous and volatile aromatic compounds, which can be of plant origin (natural) or synthetic. The crucial vital oils in broiler production are trans-cinnamaldehyde, thymol, eugenol, and carvacrol. They interfere with the bacterial enzymatic system and modulate inflammation and immune responses. They are excellent alternatives to growth-promoting antibiotics such as Avilamycin for increasing chicken production [109,110,111]. Essential oils also have a significant role in preventing and controlling necrotic enteritis in chickens [112]. Furthermore, it was reported that supplementing with essential oils such as oregano (*Origanum* genus) in broiler feed at 300–600 g/kg increased the average daily gain of broiler chickens [110]. Therefore, essential oils should be used as an alternative to antibiotics in the poultry industry.

#### 5.7.5. Nanoparticles as Feed Additives

Nanoparticles (NPs) used as supplements in broiler chicken feed are favorable for optimizing the overall health and FCR. Some NPs have been used in poultry feed to decrease harmful bacteria and stimulate beneficial bacteria in the chicken microbiota [113]. The impact of NPs on birds has been studied worldwide to better understand the overall health and growth performance, immune response, and antibacterial prospects [114]. Gangadoo et al. [114] noted that the most commonly used silver nanoparticles in commercial chicken feed were shown to boost chickens’ microbiota. However, silver nanoparticles seem to have a somewhat elevated toxicity in birds. To avoid the toxic side effects, scientists have investigated other nanoparticles that can safely contribute to the positive impact on health and productivity. A study showed that selenium NPs at concentrations ranging from 0.15 to 1.20 ppm could enhance broilers’ average daily gains [115]. Others showed that selenium NPs had a positive influence on chicken bodies, with reduced toxicity achieved at low concentrations from 0.3 to 0.5 ppm [116,117,118,119,120]. Meanwhile, similar concentrations of copper NPs as a supplement in poultry feed could have positive or negative health consequences. For example, a concentration of 50 ppm of copper at sizes 2–15 nm produced a decreased metabolic rate, oxygen consumption, and heat production and resulted in reduced embryonic development [121], whereas others showed that including copper NPs in poultry feed could boost breast and leg muscle growth, along with decreased mortality in broiler chicks [121]. Various other metals have been synthesized as NPs as supplements to poultry feed. For example, studies using metal NPs such as zinc oxide, zirconium dioxide, and platinum showed that they had a positive influence on the antibacterial activity against *Salmonella* spp. and *Staphylococcus aureus* [122,123,124]. On the other hand, Zn-bearing zeolite clinoptilolite NPs were shown to improve broilers’ growth and feed intake ratio [125]. In addition, NPs combined with extracts of natural antibacterial products, such as certain herbs and oils, could be utilized to upgrade the overall performance of broiler feed. For instance, a study using a turmeric extract in nanocapsules showed improved meat quality without affecting broiler performance.

Nanosuspensions of clay minerals improved the antibody protection against a number of infectious diseases in chickens, including Newcastle disease, infectious bronchitis, and infectious bursal disease [126]. However, more research is needed on the benefits of using nanoparticles with poultry feed as an alternative to antibiotics.

## 6. Research Gap, Status, and Future Trends

The global poultry industry has dramatically expanded to meet the increasing need for animal proteins for human consumption. The Food and Agriculture Organization projected that annual meat production by 2050 will be 200 million tons due to the global need [127]. Therefore, there is an urgent need to find strategies by conducting relevant research to improve the growth rate, feed efficiency, and health status and reduce the pathogen burden to meet the increased demand. It is possible to use feed formulations to improve the FCR, resulting in lower feed requirements to attain market weight. Feed additives such as vitamins and minerals are usually combined with typical diets to support a rapid growth and favorable FCR. Recently, various nanoparticles have been added to feed to improve the overall health performance and FCR. More studies determining the influence of NPs on the intestinal microbiota of chickens would allow scientists to understand their role in the production of beneficial gut microbes and metabolites. Recent advances in sequencing technology could allow a much-needed understanding of the interactions between NPs and intestinal bacteria. Further research that includes the toxicology, histology, and the residual effects of NPs on broiler weight and egg production is warranted before any wide adoption of NPs in feed could occur [128].

In contrast, the global consumption of antimicrobials in food animal production is anticipated to increase by two-thirds by 2030 due to rises in production and the demand for animal products. [129] Although most antibiotic use occurs in the agricultural sector, relatively little research has been done on how antibiotic use in food animals contributes to the overall problem of antibiotic resistance. There is a notable lack of quantitative monitoring data on antibiotic use in broilers in most large poultry-producing countries [40]. An increase in the progression and spread of antibiotic resistance has become a significant concern. Over the past few decades, no new major types of antibiotics have been developed and almost all known antibiotics are increasingly losing their curative sensitivity against pathogenic microorganisms. A single health approach that highlights the connections among humans, animals, and the environment is required to tackle antibiotic resistance in an integrated manner. This viewpoint entails a pledge of cooperation among professionals in several disciplines, including physicians, veterinarians, food safety practitioners, and environmental health experts.

## 7. Conclusions

Antibiotic-free poultry meat production is an old issue in developed countries; however, it is at the beginning stage in Bangladesh. Although consumers are becoming more interested in antibiotic-free poultry, it is not easy to produce quality antibiotic-free meat while maintaining animal welfare and making farming profitable. There is still a long way to go for sustainable antibiotic-free broiler meat production, particularly in developing countries like Bangladesh. The aspects related to broiler meat production and marketing should be ensured through improved management systems, proper legislation, and implementation. It is difficult to achieve sustainable production if there is any interruption of the supply chain, such as the supply of day-old chicks from hatcheries or feed from feed mills, or other support. As such, producing sustainable broiler meat without antibiotics is an ongoing challenge for the food industry in Bangladesh. To overcome this challenge, good management practices, including strict biosecurity measures, should be ascertained. Moreover, access to antibiotics without a prescription by veterinarians should be strictly enforced. In addition, for fighting against antimicrobial resistance, awareness-building programs as recommended by the World Organisation for Animal Health (OIE) (https://oie-antimicrobial.com/) should be implemented for raising awareness among poultry farm workers, marginal poultry farmers, and poultry consumers to highlight the dangers of antibiotic resistance and the benefits of consuming antibiotic-free poultry meat. As policymakers and local poultry industries are aiming to produce safe meat without using antibiotics by 2024, alternative approaches to antibiotics and active participation by farmers and consumers will be required to achieve this.

## Figures and Tables

**Figure 1 biology-09-00411-f001:**
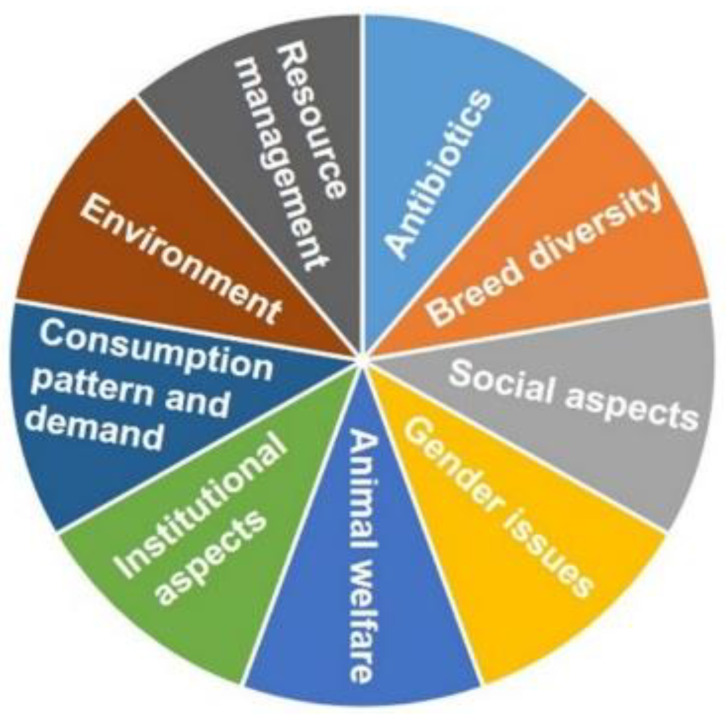
Schematic presentation of the challenges in sustainable antibiotic-free broiler production.

**Figure 2 biology-09-00411-f002:**
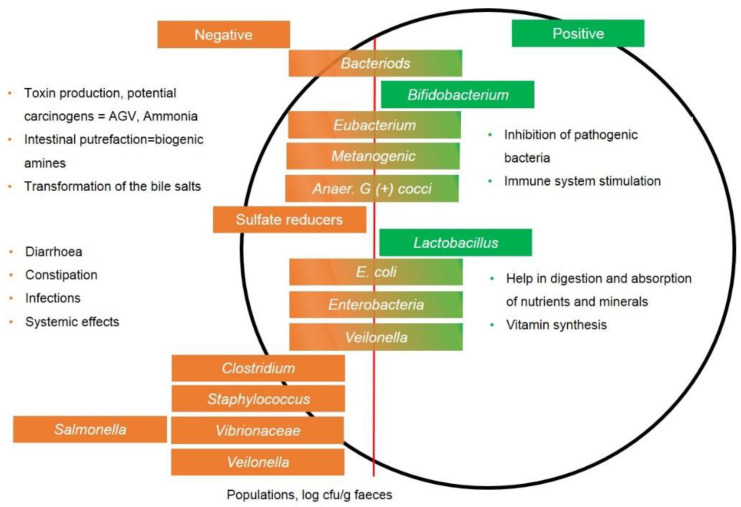
Role of gut microbial communities in the gastrointestinal tract (GIT). Green indicates the beneficial effects, and orange indicates the harmful effects of GIT bacteria. E. coli, *Escherichia coli.*

**Table 1 biology-09-00411-t001:** General approaches for the production of antibiotic-free broilers.

Strategy	Specific Action
Apply minimum 14 day interval for new flock entry	Reduce the frequency of pathogens
Treat feed to reduce bacterial pathogens	Maintain bacterial CFU (colony-forming unit) to <10 after converting finely ground mash feed to capsule or <10^3^ CFU at the farm
Process vegetable formulated feed in diet	Obtain low fat and high protein and reduce the likelihood of clostridia contamination
Maintain dry litter	Reduce ammonia level and stocking thickness, improve ventilation, increase the distance of shavings, etc.
Apply good sanitation program for drinking water	Reduce contamination of bacteria and remove biofilm from the pipelines, regulators, and nipple drinkers
Dispose of dead birds regularly	Prevent cannibalism and minimize bacterial contamination
Use probiotic supplements for early stages of broiler feed	Establish healthy gut microorganisms and increase growth performance
Grind coarser grain to finest	Upgrade the role of gizzard and digestion
Supplement with whole grain or grits	Reduce the temporary loss of growth rate and feed efficiency and progress the function of gizzard and digestion
Add essential oil extracts to feed	Maintain bacteria at safe levels and improve intestinal health
Reduce nonprotein nitrogen by preparing feeds based on digestible amino acids	Inhibit the proliferation of bacteria
Utilize ingredients with more soluble fiber	Avoid the deposit of insoluble fiber in the hindgut
Use digestible fats and starches	Help suitable digestion, prevent non-starch polysaccharides from getting into the hindgut
Lessen the addition of ingredients like wheat, barley, and oats	Minimize gut damage and subsequent enteritis
Maintain proper electrolyte balance	Decrease flushing and feed passage
Limit feed changes	Reduce disturbances of the gut microflora
Add exogenous enzymes	Exploit extraction and digestion of nutrients and minimize the viscosity of digesta
Maintain good management practices	Minimize stress
Follow good biosecurity practices	Reduce the opportunity for disease

**Table 2 biology-09-00411-t002:** Antibiotic alternatives and their functions and effects in broiler production. FCR, feed conversion ratio and BW, body weight.

Antibiotic Alternatives	Active Ingredients		Basic Functions	Advantages	Disadvantages	Effects on Broiler	References
Probiotics	*Bacillus subtilis*, *Enterococcus faecium*, *Lactobacillus acidophilus*, *Bacillus licheniformis*, *Bifidobacterium bifidum*		Appetite and digestion, stimulant, antioxidant	Modulation of immunity Proliferation of beneficial bacteria Increased nutrient absorption No development of resistance Stable	No antibacterial properties	Increased body weight and FCR Improved absorptive surface of duodenum and ileum Increase nutrient retention	Ghasemi et al. [72]; Giannenas et al. [73]; Levkut et al. [76]; Latha et al. [77]; Zhang et al. [78]; Popova [79]; Liu et al. [80]
Prebiotics	Fructo-oligosaccharides (FOS), inulin, galacto-oligosaccharides (GOS), trans-galacto-oligosaccharides (TOS)		Digestion, stimulant	Modulation of immunity Proliferation of beneficial bacteria No development of resistance Stable	No antibacterial properties Unknown nutrient absorption	Increased growth performance Stimulation of metabolic activity in intestine	Jozefiak et al. [82]; Morales-Lopez et al. [83]; Zhang et al. [84]; Baurhoo et al. [86]; Baurhoo et al. [87]
Organic acids	Citric acid		Digestion, stimulant, increased feed efficiency	Antibacterial properties Modulation of immunity Increased nutrient absorption Stable	Development of resistance is rare	Increased body weight Improved ileal nutrient digestibility, cell proliferation, and epithelial and villi height	Kum et al. [88]; Hassan et al. [90]; Nava et al. [91]; Adil et al. [92]; Chaveerach et al. [93]; Izat et al. [94]; Mohammadagheri et al. [95]; Hu and Guo [97]; Qaisrani et al. [98]
	Ascorbic acid						
	Propionic acid and sodium bentonite						
	Butyrate						
Amino acids and enzymes	Phytase, lysins		Digestion, stimulant	Antibacterial properties Modulation of immunity		Improved growth performance	Cowieson et al. [99]; Fenton et al. [100]; Rios et al. [101]; Volozhantsev et al. [102]
Phytogenic feed additives	Pepper	Piperine	Digestion, stimulant	Modulation of immunity Antibacterial properties Proliferation of beneficial bacteria Increased nutrient absorption No development of resistance Stable		No effect on live performance	Windisch et al. [103]; Frankic et al. [104]; Toghyani et al. [105]; Li et al. [106]; Mpofu et al. [107]; Kirubakaran et al. [108]; Khattak et al. [109]; Peng et al. [110]; Pirgozliev et al. [111]; Jerzsele et al. [112]
	Garlic	Allicin	Digestion, stimulant, antiseptic			Higher body weight	
	Ginger	Zingerone	Gastric stimulant			No effects on performance	
	Rosemary	Cineol	Digestion, stimulant, antiseptic, antioxidant			Improved live weight and feed efficiency	
	Thyme	Thymol	Digestion, stimulant, antiseptic, antioxidant			No significant effect on BW/FCR	
	Mint	Menthol	Appetite, digestion, stimulant, antiseptic			Decreased serum total cholesterol, triglycerides, and low-density lipoprotein concentration	
Nanoparticles (NPs)	Silver NPs		Digestion, Stimulant	Modulation of immunity Antibacterial properties Proliferation of beneficial bacteria Increased nutrient absorption No development of resistance Stable	Some toxicity in broilers	Increased body weight and FCR	Gangadoo et al. [114]
	Selenium NPs						Fuxiang et al. [115]; Hu et al. [116]; Cai et al. [117]; Mohapatra et al. [118]; Bagheri et al. [119]; Selim et al. [120]
	Copper NPs						Gangadoo et al. [114]; Mroczek-Sosnowska et al. [121]
	Metal NPs such as zinc oxide, zirconium dioxide, and platinum						Akbar and Anal [122]; Ravikumar and Gokulakrishnan [123]; Prasek et al. [124]
	Zn-bearing zeolite clinoptilolite NPs						Tang et al. [125]
	Nanosuspensions of clay minerals						Elshuraydeh [126]

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
