# Peer review of "Sustainable Antibiotic-Free Broiler Meat Production: Current Trends, Challenges, and Possibilities in a Developing Country Perspective"

_biology, 2020, doi:10.3390/biology9110411_

Round 1

Reviewer 1 Report

In this review it is shown that chickens are raised with the assistance of regular uses of antibiotics not only for the prevention and treatment of diseases but also for body growth. Therefore, antibiotic-free broiler meat production is becoming increasingly popular worldwide to meet consumer demand. This review focuses on the current scenario of antibiotic use, prospects, and challenges toward sustainable antibiotic-free broiler meat production

 It is a very interesting work which is very well structured and reasoned. It contains very important information that must be known.

The percentage of each of the critical challeges should appear in Figure 1

The second figure has no identification either on the manuscript or on the figure's foot

Reviewer 2 Report

General comment

The work is well organized, and the topics are adequately presented. The language is good and only slight corrections would be required.

Specific concerns:

Line 53: "…raising animals with the antibiotic is a challenge." Probably is "without the antibiotic".

Lines 91-259-274-364-547: Conform the references in all the text according the guidelines for authors. i.e. "Sattar et al. 2104" replace with "Sattar et al. [14] ".

Line 116: Replace "percent" with "%".

Line 117: Delete the sentence: "It is also ensuring food security".

Line 119-120: Delete the sentence: "It is creating procuring authority and plummeting shortage on a high level".

Lines 120-125: Conform the numbers.

Line 173: Delete "Bardhan]".

Line 396: Replace "&" with "and".

Line 450: Insert the reference numbers at the end of the sentence.

Line 456: Write the genera in cursive font.

Line 523-544-585: For "feed conversion ratio (FCR)" use the abbreviation after the first time.

Line 524: Use the abbreviation "PFAs" for "phytogenic feed additives".

Line 537: Correct the reference numbers [109,110,111] according the guidelines.

Table 2. Replace "Silver nanoparticles" with "NPs".

Reviewer 3 Report

Title: Sustainable antibiotic-free broiler meat production: Current trends, global challenges, and possibilities

After looking at the text and highlighting some issues, it became too much to correct the English, so the identification of issues was discontinued.

While antibiotics and sustainability are related, the plethora of issues seems to recommend that they be covered in separate papers. The current paper’s coverage of antibiotics is mostly a summary one country’s experiences. The authors admit that there is lawless use of antibiotics. Thus, the major problem with antibiotic use is the development and enforcement of institutions. Ideas for relating antibiotics and sustainability cannot be meaningfully addressed in societies that cannot or do not have the ability to enforce needed rules. This means that the paper is simply an academic exercise without application in many areas of the world.

It is not clear that a paper can reasonably cover the issues of antibiotics and sustainability without differentiating between developed countries and developing countries. The issues are too different. As written, the paper seems to mainly concern developing countries, so this might need to be acknowledged in the title and the paper.

From the sustainable part of the paper, there is not anything about preserving sufficient resources for future generations.

The conclusion of the paper seems to give up on relating antibiotics to sustainability. It first addresses sustainability, and closes with antibiotics. A relationship between them is not established. This suggests the review is not successful and is not worthy of publication.

1. L53 “However, raising animals with the antibiotic is a challenge, . . “ - shouldn’t “with” be “without.”
2. L59 “broiler making” needed to be changed as producers do not make broilers.
3. L60-62 “In addition, consumers are concerned about antibiotic residue, antimicrobial resistance, among other things, such as pesticide residues, additives, nutritional content, flavor, traceability, regional production, genetically modified organism, social justice, etc.” – not clear how these relate, if at all, with antibiotic use.
4. L66-68 The text before “[3]” seems to require two sentences or rewriting.
5. L72-73 Doesn’t any antibiotic use favor resistance?
6. L82 “antibiotic-resistant” should be “antibiotic-resistance.”
7. L88 “The cost-effective way” needs to be reworded.
8. L91 “Sattar et al. 2104" should be 2014?
9. L94 What “easily available, but illegal, action” is being discussed?
10. L100 Are antibiotics “leading to severe public health ailments”? Which ailment?
11. L119+ What is meant by “It is creating procuring authority and plummeting shortage on a high level”?
12. L122 Revise “the country investment; a Taka 200 billion industry.”
13. L145+ This should be removed from the paper as it is no longer true. The US figure is from data collected before 2010). The FDA subsequently adopted a VFD that has drastically cut antibiotic use in poultry. More than one-half of poultry products sold in the US came from birds that were not fed antibiotics.
14. L150 A review paper should not be concerned with figures from a country that is not among the world’s largest poultry producers.
15. L166-67 Readers will not understand “to 1,022 crore pieces and 14.6 lakh tons, respectively. Thus, the poultry industry is worth about Tk 30,000 crore.”
16. L169 How can there be “there is a large market for antibiotic-free chickens in Bangladesh” if there is no testing and many producers are using antibiotics?
17. L429 “Removal of feed antibiotics will have no significant effects on broiler production despite an initial increase in cost to diseases.” So why are producers using antibiotics? Moreover, in the conclusion the opposite is stated.
18. L430 What are “DOCs”?
19. L580 Given earlier cited illegal use of antibiotics, some might disagree that “primary research should focus on upgrading productivity.” Another conclusion is that primary research should be how to have producers comply with laws.
20. L596 A citation is needed for “There is an enormous lack of quantitative monitoring data on antibiotic usage in broilers in most large poultry-producing countries.”
21. L610+ The paper gives up on its discussion and application to developed countries.
22. L612+ “is not easy to produce antibiotic-free quality broiler meat maintaining animal welfare issues and making the farming profitable.” Producers of over one-half of the US poultry products seem to have worked this out.

Round 2

Reviewer 3 Report

The authors have addressed most of the individual comments previously identified by line numbers, but ignored the general comments.

The title is unacceptable.

Many of the grammatical English problems have not been corrected. The text does not seem to meet the standards of this journal.

The current paper’s coverage of antibiotics is mostly a summary one country’s experiences. This is stated in sections 2, 3 and 4. The topic of the paper needs to be reconciled with the title. The review does not highlight the current scenario of antibiotics uses in broiler meat production as it ignores most of the poultry production around the world.

The authors have added a new reference, 42. The FDA reports that only 6% of medically important antimicrobials were for chickens. This figure seems to be at odds that widespread use of antibiotics in broiler farms may be causing a global public health threat. Moreover, reference 43 addresses antibiotic use 15 years ago in the U.S. The FDA subsequently adopted a VFD that has drastically cut antibiotic use in poultry. More than one-half of poultry products sold in the US came from birds that were not fed any antibiotics.

The authors claim that most poultry meat consumers avoid broiler meat, fearing potential
residual antibiotics, and antimicrobial resistance. World-wide data on poultry consumption show an increase in sales. It is the favorite meat in Japan and the U.S.

The paper does not cover the issues of antibiotics and sustainability in a meaningful manner. It looks at one country and addresses factors of sustainable production without tying the factors to practices.

From the sustainable part of the paper, there is not anything about preserving sufficient resources for future generations.

The authors did not revise their conclusion. The paper seems to give up on relating antibiotics to sustainability. It first addresses sustainability, and closes with antibiotics. A relationship between them is not established. This suggests the review is not successful and is not worthy of publication.

Round 3

Reviewer 3 Report

none